# Extraction Optimization, Structural Characterization, and Anticoagulant Activity of Acidic Polysaccharides from *Auricularia auricula-judae*

**DOI:** 10.3390/molecules25030710

**Published:** 2020-02-06

**Authors:** Chun Bian, Zhenyu Wang, John Shi

**Affiliations:** 1School of Chemistry and Chemical Engineering, Harbin Institute of Technology, Harbin 150001, China; ji3526183@126.com; 2School of Food Engineering, Harbin University, Harbin 150086, China; 3Agriculture and Agri-Food Canada Guelph Food Research Center, Guelph, ON N1G5C9, Canada

**Keywords:** corresponding surface optimization, ultrasound assisted extraction, chemical composition, anticoagulation in vitro

## Abstract

To explore *Auricularia auricula-judae* polysaccharides (AAP) as natural anticoagulants for application in the functional food industry, ultrasound assisted extraction (UAE) was optimized for the extraction of AAP by using a response surface methodology (RSM). The maximum extraction yield of crude AAP (14.74 mg/g) was obtained at the optimized extraction parameters as follows: Extraction temperature (74 °C), extraction time (27 min), the ratio of liquid to raw material (103 mL/g), and ultrasound power (198 W). Furthermore, the acidic AAP (aAAP) was precipitated with cetyltrimethylammonium bromide (CTAB) from crude AAP (cAAP). aAAP was further purified using ion exchange chromatography with a DEAE Purose 6 Fast Flow column to obtain aAAP-1. Additionally, according to the HPLC analysis, the aAAP-1 was mainly composed of mannose, glucuronic acid, glucose, galactose, and xylose, with a molar ratio of 80.63:9.88:2.25:1:31.13. Moreover, the results of the activated partial thromboplastin time (APTT), prothrombin time (PT), and thrombin time (TT) indicated aAAP-1 had anticoagulant activity, which was a synergic anticoagulant activity by the endogenous and exogenous pathway.

## 1. Introduction

A hypercoagulable state or thrombus can be induced by an imbalance between coagulation and anticoagulation or coagulation and fibrinolysis, which also leads to the instability of atherosclerosis plaque, thus inducing ischemic cardiovascular and cerebrovascular disease [1]. According to the World Health Organization (WHO), thrombosis-related cerebrovascular disease and stroke are the main factors of mortality worldwide, which are expected to cause almost 3.6 million deaths by 2030 [2]. Anticoagulants are widely used in the medicinal field for treating diseases like strokes and myocardial infarction [3]. Since the 1940s, heparin has been the predominant anticoagulant and antithrombotic drug [4]. However recently, it has been found that heparin has several side effects, such as bleeding, thrombocytopenia [5,6], hyperkalemia, and risk of infection with animal pathogens, due to its origin [7,8]. Hence, it is necessary to find a safe anticoagulant from a natural substitute for heparin. It has been reported that polysaccharides with anticoagulant and antithrombotic activity are not only from marine origins with a sulfated structure [3,4,9] but also are from plants and edible fungi [10,11].

*A. auricula-judae* is a kind of saprophyte that grows on deadwood sourced as either wild or cultivated and its subsidiary is the edible part. It is the main genus among edible macro fungi, and northeast China is the main production area [12]. *A. auricula-judae* is a species of food and medicine homology particularly used in traditional Asian medicine. *A. auricula-judae* is rich in proteins, fats, vitamins, and trace elements [13], besides, its polysaccharides (AAP) are the main functional component [14]. Many biological functions of AAP have been reported by previous studies, such as antioxidant activity [15,16], immunomodulatory activity [17], hypoglycemic or anti-diabetic properties [18], radio-protective properties [19], anti-cancer or anti-tumor properties [20,21], anti-bacterial [22] or anti-viral properties [23], and anti-hypercholesterolemic properties [24]. In addition, AAP also has an anticoagulant function due to the structure of uronic acid [25]. The cAAP can be extracted through alkali-soluble alcohol precipitation, and further purified by gel chromatography to get the acid AAP (aAAP). The specific anticoagulant activity of aAAP was 2 IU/mg when its average mass was ~160 kDa, and the anticoagulant activity of AAP was mainly contributed by glucuronic acid [26] but not sulfate ester, which is the anticoagulant active ingredient of heparin [27].

Ultrasound-assisted extraction (UAE) uses ultrasound waves with a frequency of 20 to 100 kHz. Ultrasound equipment can be divided into an ultrasonic bath (indirect sonification, 45–50 kHz) and ultrasonic probe (direct sonification, 20 kHz) [28]. The costs of the equipment are lower than other alternative extraction techniques, and a wide variety of solvents (water, ethyl alcohol, and other organic solvents) can be used [29,30]. UAE can notably shorten the extraction time, lower the extraction temperature, through cavitation bubbles, cell disruption, and particle size reduction, and enhance the contact between targeted compounds and solvents [31,32]. Under the above principles, UAE can also reduce the solution viscosity [14], increase the biological functions, and affect the structural composition of the extracted substance [28,33]. Pawlaczyk-Graja et al. [34] compared different methods, including cold extraction, hot extraction, and microwave-assisted extraction (MAE), and UAE. All the methods were carried out in 0.1M NaOH and under their respective best parameters to obtain polyphenolic–polysaccharide conjugates with anticoagulant activity from the leaf of *Fragariavesca* L. UAE gave rise to significant differences in the polysaccharide structures of the conjugates, increased the extraction yield, and the extracts with the highest anticoagulant activity were compared the other two methods aforementioned. Lin et al. [35] reported that the polysaccharides extracted from *Ziziphus jujuba* Mill var. *spinosa* seeds had higher antioxidant activity and immunoregulation through UAE than those extracted by heating water extraction (HWE), but the yield was similar, which roughly equated to 0.93 ± 0.14% of 6 h HWE. Yip et al. [36] reported that the yield of polysaccharides was higher, extracted from *Ginseng radix* and Dendrobii Officinalis Caulis by UAE, compared to boiling water extraction (BWE). The powerful extraction ability and polysaccharide degradation caused by ultrasound collectively contributed to these differences. Optimal extraction was required for the polysaccharides of different sources [36]. In order to explore AAP as functional food ingredients and natural anticoagulants for industrial applications, the UAE method was first optimized to extract crude AAP by using RSM. The crude AAP was further purified, and the chemical structures and anticoagulant activities of AAP obtained were investigated.

## 2. Results and Discussion

### 2.1. Single Factor Tests of UAE

The ultrasound extraction temperature, extraction time, ratio of liquid to raw material, and ultrasonic power are significant parameters that affect the extraction yield of polysaccharides [14,32,37]. The effects of these parameters on the yields of AAP are shown in Figure 1. Briefly, Figure 1A shows that the yield of AAP increased slowly with increasing temperature, eventually reaching a maximum at 70 °C, and then the yield decreased. As the temperature increased, degradation of polysaccharides resulted in a reduction of the surface tension and the viscosity of the solvent [38], increasing the steam pressure in small bubbles, and thus decreasing the ultrasonic cavitation and the mass-transfer intensity [39]. Eventually, the extraction yield dropped. The yields of AAP increased, while the extraction time increased from 10 to 25 min, and the highest extraction yield was reached at 25 min (Figure 1B). However, a longer extraction time induced the degradation of polysaccharides, hence the yield decreased when the extraction time increased continuously [40]. In addition, the yields of AAP were improved as the ratio of liquid to raw material increasing from 30 to 100 mL/g, reaching the highest yield at 100 mL/g (Figure 1C). However, the AAP yields decreased slowly while the ratio of liquid to raw material continued to increase. This might be due to a lower density and viscosity caused by a higher ratio of liquid to raw material, facilitating the dilution of polysaccharides in the solvent [41]. Furthermore, AAP yields were positively related to ultrasound power from 80 to 225 W, whereas the yields were negatively related to the power after 225 W (Figure 1D). The application of ultrasonic energy produces instantaneous high pressures and temperatures, which promote the formation of cavitation bubbles. Therefore, the solvent more easily destroys the cell wall and enhances the release of the components. However, the excessive sonication energy input exceeds the energy required for cracking, resulting in structural changes and degradation [30]. Finally, the optimal extraction conditions were investigated as 70 °C, 25 min, 100 mL/g, and 225 W.

### 2.2. Optimization of the Yield of AAP by RSM

Based on the results of the single factor tests, 29 runs of Box–Behnken experimental design (BBD) were utilized to optimize the UAE conditions. The coded values and the yields of APPs under different extraction conditions are shown in Table 1, and the AAPs yields varied from 9.04 to 16.17 mg/g. All the data were analyzed through multiple regression analysis; a second-order polynomial equation for AAPs yield is represented as Equation (1):*y* = 15.52 + 0.40*X*_1_ + 0.36*X*_2_ + 0.40*X*_3_ + 0.18*X*_4_ − 0.87*X*_1_*X*_2_ − 0.64*X*_1_*X*_3_ − 0.74*X*_1_*X*_2_ − 0.91*X*_2_*X*_3_ − 1.19*X*_2_*X*_4_ − 1.45*X*_3_*X*_4 −_ 2.34*X*_1_^2^ − 2.05*X*_2_^2^ − 2.33*X*_3_^2^ − 2.15*X*_4_^2^,(1)
where *y* represents the predicted yield of AAP, and *X*_1_, *X*_2_, *X*_3,_ and *X*_4_ are the extraction temperature, extraction time, ratio of liquid to raw material, and ultrasonic power, respectively.

The analysis of variance (ANOVA) for the fitted quadratic polynomial model of AAP extraction is shown in Table 2. A high model *F*-value (40.80) and a very low *p*-value (<0.0001) indicated that the mode of AAP yield was very significant. Meanwhile, the lack of fit of the *F*-value (0.71) and *p*-value (0.6989) implied that it was insignificant relative to the pure error, which confirmed the goodness-of-fit and suitability of the model for the prediction of the response values under any combination of the independent variables [36].

The determination coefficient, *R*^2^ (0.9761), and the adjusted determination coefficient, *R*^2^ (0.9522), were reasonably close to 1. The difference between the two coefficients was within 0.2, indicating a high degree of correlation between the observed and predicted values [36]. Furthermore, the low values of the coefficient variation (CV, 3.73%) and the high values of adequate precision (15.024) revealed the high degree of precision and reliability of the observed values. The linear coefficients (*X*_1_, *X*_2_, and *X*_3_), interaction coefficients (*X*_1_*X*_2_, *X*_1_*X*_3_, *X*_1_*X*_4_, *X*_2_*X*_3_, *X*_2_*X*_4_, and *X*_3_*X*_4_), and quadratic coefficients (*X*_1_^2^, *X*_2_^2^, *X*_3_^2^, and *X*_4_^2^) were all significant at the level of *p* < 0.05 or *p* < 0.01. The other term coefficients were not significant (*p* > 0.05). After removing the insignificant factor, the final empirical model could be described as Equation (2):*y* = 15.52 + 0.40*X*_1_ + 0.36*X*_2_ + 0.40*X*_3_ − 0.87*X*_1_*X*_2_ − 0.64*X*_1_*X*_3_ − 0.74*X*_1_*X*_2_ − 0.91*X*_2_*X*_3_ − 1.19*X*_2_*X*_4 −_ 1.45*X*_3_*X*_4 −_ 2.34*X*_1_^2^ − 2.05*X*_2_^2^ − 2.33*X*_3_^2^ − 2.15*X*_4_^2^.(2)

The predicted values obtained from the model were in agreement with the experimental result (Figure 2A). Meanwhile, the normal probability plots of residuals indicate no serious deviation from normality (Figure 2B) [39].

The predicted models are presented in contour plots (Figure 3). Figure 3A−F depict that the interactions between any two extraction factors in this experiment were both significant, and the interactions of these factors on the efficiency of AAP extraction were an additive effect [42,43]. Furthermore, the model predicted that the maximum yield (14.74 mg/g) could be obtained by the optimal extraction conditions: Extraction temperature of 74.12 °C; extraction time of 27.28 min; liquid to solid ratio of 102.57 mL/g; and microwave power of 198.79 W. Considering the practical operability, the operating conditions were: Temperature of 74 °C, time of 28.0 min, ratio of liquid to raw material of 100 mL/g, and ultrasonic power of 200 W. The actual yield was 14.07 ± 0.21 mg/g (*n* = 3), which was similar to the predicted value. Therefore, the model of UAE was accurate and adequate in this study. The AAP was prepared by using the above optimized extraction scheme.

### 2.3. Purification and Monosaccharide Composition of aAAP-1

After the removed protein was concentrated, dialyzed, and lyophilized, the yield ratio of cAAP was about 80.68%. aAAP was precipitated from cAAP with CTAB and purified by using ion exchange chromatography with DEAE Purose 6 Fast Flow column (Figure 4). The yield ratio of aAAP was 62.71%. The yield ratios of aAAP-2 and aAAP-3 were lower than that of aAAP-1 (15.16%). Only aAAP-1 was further studied.

As shown in Figure 5, aAAP-1 exhibited a single and symmetric peak in the HPLC chromatography, indicating that aAAP-1 was a homogeneous polysaccharide. According to the regression equations (lg*M*_w_ = −0.1973*t* + 12.456, *R*^2^ = 0.9957; lg*M*_n_ = −0.1776*t* + 11.527, *R*^2^ = 0.9966), the *M*_w_ and *M*_n_ of aAAP-1 were 1538 and 1525 Da, respectively, with a retention time of 46.98 min, and *M*_w_/ *M*_n_=1.008, which represented the high homogeneity of aAAP-1. 

The HPLC chromatographs of monosaccharide standard and aAAP-1 are shown in Figure 6. These show that aAAP-1 was an acidic heteropolysaccharide mainly composed of Man, GlcUA, Glc, Gal, and Xyl, with molar ratio of 80.63:9.88:2.25:1:31.13. The aldehyde acid of aAAP-1 was glucuronic acid but not galacturonic acid [44], which is consistent with Zhang et al. [45]. The monosaccharide composition and molar ratio of aAAP-1 was different from other polysaccharides extracted from *A. auricula-judae* [26,46], which indicated that the differences in the extracting conditions, separation, and purification influence the composition of polysaccharides from the same resource.

### 2.4. UV and FT-IR Spectrum of aAAP-1

The ultraviolet-visible (UV-vis) spectrum of aAAP-1 is shown in Figure 7A, showing only an absorption peak at 200 nm and without absorption peaks at 260 and 280 nm, indicating that the aAAP-1 did not contain nucleic acids and proteins [47]. The FT-IR spectrum of aAAP-1 was scanned in the range 400–4000 cm^−1^ and is presented in Figure 7B. A weak peak near 3730 cm^−1^ indicated a free intermolecular hydroxyl (–OH), the broad and strong absorption peak at 3440 cm^−1^ (3500–3100 cm^−1^), which could be assigned to stretching vibrations of intramolecular hydroxyl (–OH) groups [48]. The absorption peaks at 2930 and 2850 cm^−1^ (2800–3000 cm^−1^) were formed by the chemical bond (C–H) stretch [49], and the peak at 2360 cm^−1^ was the C–H transiting angle [50]. These are typical characteristics of polysaccharides. Uronic acids were evidenced by the strong absorbance band at 1650 cm^-1^ (C–O) and two bands at 1480 and 1380 cm^−1^ (O–C=O) [3,51]. In addition, the absorption peaks around 1070 cm^−1^ (1150-1050 cm^−1^) were ascribed to C–O–C and C–O–H vibration, indicating the existence of pyranose. The characteristic absorption bands at 841 cm^−1^ suggested the existence of β-configurations of saccharide units in aAAP-1 [39]. The S=O stretching is characterized by no absorbance band at 1240 cm^-1^, proving that aAAp-1 is not a sulfate ester [26].

### 2.5. Anticoagulant Activity of aAAP-1 In Vitro

Thrombosis and other blood coagulation with thrombus are caused by a disorder of the balance between the coagulant and anticoagulant system in the body. The activation of the anticoagulation system is an important mechanism for antithrombosis [52]. With a series of enzymatic reactions, the coagulation process in the body can be divided into three stages: Formation of thrombin activator, activation of prothrombin, and transformation of fibrinogen into fibrin. Meanwhile, there are also three accurately controlled anticoagulant systems in the body’s blood: 1. Serine protease inhibitors, such as plasminogen activator inhibitor; 2. the protein C/S0 anticoagulant system; and 3. Surface-binding inhibitor and coagulation inhibitor [53]. Initiated by different mechanisms, the coagulation process can be divided into an exogenous and endogenous coagulation pathway.

The experiments of APTT, PT, and TT were used to test the coagulation time, and estimate the anticoagulant mechanism and pathway. APTT is a test of the endogenous clotting activity. AAAP-1 (47.3 s, 50.0 μg/mL) prolonged the clotting time of APTT significantly compared to the saline control (28.0 s), but it was much lower than that of heparin (48.7 s, 2.0 μg/mL). This result is very close to the studies of Li, C. et al. [54] and Lin, S. et al. [55], and represented a deficiency in factors VIII, IX, XI, XII, or Von Willebrand’s factor (VWF). AAAP-1 in vitro (12.5–50 μg/mL) could prolong PT (16.1–73.7 s) and TT (20.3–28.2 s) significantly (*p* < 0.01) compared to the saline control (11.3 and 17.7 s). However, these values were also much lower than those of heparin (67.9 and 32.7 s, 2.0 μg/mL). Additionally, with the increasing dose, PT and TT were lengthened, exhibiting a dose–effect relationship (Table 3). As an important index, TT is mainly a reflection of the degree of the conversion of fibrinogen into fibrin [56]. The clotting time of TT was prolonged significantly, which indicated that aAAP-1 inhibited thrombin activity, and a previous study observed this anticoagulant activity is mostly mediated by the catalysis of thrombin inhibition by antithrombin but not by heparin cofactor II [26]. PT was used to evaluate the overall efficiency of the exogenous clotting pathway. A prolonged PT indicated a deficiency in coagulation factors V, VII, and X. In conclusion, aAAP-1 had a synergic anticoagulant activity by the endogenous and exogenous pathway. Moreover, the anticoagulant activities of aAAP-1 were weaker than those of heparin.

## 3. Materials and Methods

### 3.1. Material and Chemicals

*A. auricula-judae* was supplied by a forest farm in the Yichun District, Yichun, Heilongjiang Province, China. Standards of Man, Rha, GlcUA, GalUA, Glc, Gal, Xyl, Ara, and Fuc were from Aladdin Chemical Reagent Co., Ltd. (Shanghai, China). Standard dextrans, CTAB, and the standard heparin were all purchased from Sigma-Aldrich Chemical Co. (St. Louis, MO, USA). DEAE Purose 6 Fast Flow was purchased from Jiansu Qianchun Biological Technology Co., Ltd. (Jiangsu, China). APTT, PT, and TT commercial kits were all purchased from Nanjing JianCheng institute of biological engineering (Nanjing, China). Normal plasma and all other reagents and chemicals used were of analytical grade.

### 3.2. Extraction and Purification of Acidic Polysaccharides from A. auricula-judae

#### 3.2.1. Ultrasound-Assisted Extraction (UAE) of AAP

UAE was performed based on a previously reported method with some modifications [39]. Both the designs of the single-factor experiment and Box–Behnken experimental (BBD) were also applied for the optimization of the UAE conditions. Briefly, after being dried to a constant weight at 70 °C, *A. auricula-judae* was ground to a particle diameter size of 20–50 μm and defatted by extracting with ethyl acetate and methanol. The removal of some colored materials was achied with a treatment with 95% EtOH following [57]. The extracted residue was extracted with sodium hydroxide solution (0.01M) by continuous UAE (HX-900 ET, Shanghai Huxi Industrial co. LTD, Shanghai, China). The effects of different extraction temperatures (40, 50, 60, 70, 80, 90, and 100 °C), different extraction times (10, 15, 20, 25, 30, 35, and 10 min), and the liquid-solid ratio (20, 40, 60, 80, 100, 120, and 140 mL/g), on the yields of AAP (mg/g) were investigated by using a single-factor experimental design. While the ultrasound power were set as 225 W, the extraction time and the liquid-solid ratio were set as 25min and 100 mL/g), the extraction temperature and the liquid-solid ratio were set as 70 °C and 100 mL/g, and the extraction temperature and the extraction time were set as 70 °C and 25 min. 

A three-level BBD was applied to further optimize the UAE conditions. The extraction temperature (*X*_1_, °C), extraction time (*X*_2_, min), liquid-solid ratio (*X*_3_, mL/g), and ultrasound power (*X*_4_, W) were preferred for the independent variables. The variables and their levels are presented in Table 4.

The NaOH solution of polysaccharides was neutralized with 0.4 M HCl, the solids were separated by centrifugation (4000 rpm, 5 min), and the polysaccharides were precipitated from the supernatant by an excess of ethanol (4:1, *v*/*v*) precipitation. Then, flowed deproteinization and dialysis (3.5 kDa) was performed, the output was freeze-dried, and the crude cAAP was obtained [10].

#### 3.2.2. Isolation and Purification of aAAP

Using twice the volume of cAAP (10 mg/mL), water solution was mixed with CTAB (5%) water solution for 4 h. The precipitation was aAAP separated by centrifugation [58]. aAAP (400 mg) was dissolved in 2 mL of distilled water, and filtrated with a syringe filter (0.22 μm). The solution was applied to an anion exchange column DEAE Purose 6 Fast Flow (Φ × h, 2.6 × 20 cm) and sequentially eluted using 0.0, 0.2, and 0.4M NaCl aqueous solution with a flow rate of 1 mL/min. The elution of the polysaccharides was 10 mL per tube and monitored at 490 nm by using the phenol-sulfuric acid method [59]. The eluate of the same fraction was pooled, concentrated, dialyzed (3.5 kDa), and lyophilized.

### 3.3. Determination of the Molecular Weight and Monosaccharide Composition Analysis

The molecular weight (*M*_w_, *M*_n_) of aAAP-1 was measured by using Shimadzu LC-10AVP HPLC system (Shimadzu Corporation, Kyoto, Japan), equipped with BRT 105-104-102 Tandem gel column (Φ × h, 8.0 mm× 300 mm, Borui Saccharide, Biotech. Co. Ltd., YangZhou, China) and a Parallax detector. After being centrifuged (12000 rpm, 10 min) and filtered with a 0.45-μm filter, the (5.0 mg/mL) supernatant was injected into the detector. The 0.05 M NaCl was regarded as the flow phase, and the elution rate was 0.6 mL/min. The standard curve, where the elution time was plotted against the logarithm of the molecular weight, was made using dextran as a standard (1152, 11,600, 23,800, 48,600, 80,900, 148,000, 273,000, and 409,800 kDa).

Monosaccharide composition analysis of aAAP-1 was performed using a Shimadzu LC-10A HPLC system (Shimadzu Corporation, Tokyo, Japan) with an Agilent eclipse plus C18 column (Φ × h, 4.5 mm × 150 mm, 5 μm, Agilent Technologies Inc., Santa Clara, CA, USA). The detection wavelength was set at 254 nm. The temperature of the column was maintained at 35 °C. The mobile phase consisted of acetonitrile (A) and NaH_2_PO_4_ buffer (B: 0.45 g NaH_2_PO_4_ + 900 mL purified water + 1.0 mL TEA + 100 mL acetonitrile) with a flow rate of 1.0 mL/min. A gradient program was conducted by 94% B + 6% A (*V*:*V*), 4 min and 88% B + 12% A(*V:V*), 55 min.

### 3.4. UV and FT-IR Analysis

AAAP-1 was dissolved in distilled water and scanned on a Perkin Elmer Lambda750 UV–vis spectrophotometer (Waltham, MA, USA) in the wavelength range from 200 to 500 nm. FT-IR spectra of −1 were recorded using a Fourier transform infrared (FT-IR) 8400S spectrometer (Shimadzu Corporation, Tokyo, Japan) by mixing the sample (2 mg) with KBr powder (100 mg) and pressing it into a disk (10 mm diameter). The FT-IR spectrum of aAAP-1 was recorded in the frequency range 400 to 4000 cm^−1^.

### 3.5. Blood Coagulation Assays

Normal human plasma was prepared from healthy donors without a history of bleeding or thrombosis. The preparation of plasma followed previous methods [12]. aAAP-1 of different concentrations (50, 25, 12.5 μg/mL) was mixed with plasma (sample: plasma=1: 4 (*v*/*v*)), followed by incubation at 37 °C for 5 min, and then the clotting times were recorded in a CA7000 automatic coagulation analyzer (Sysmex Corporation, Kobe, Japan). The samples, including the positive control of heparin (2 μg /mL), were dissolved in saline, and a control group of saline only was used.

### 3.6. Statistical Analysis

All experiments were conducted in triplicate, and data were expressed in means ± standard deviations. The obtained data were analyzed by the statistical package of the Design Expert software 8.0.5 (Stat-Ease Inc., Minneapolis, MN, USA). The independent sample t-test (*p* < 0.05) was used to explore significant differences.

## 4. Conclusions

In this study, the optimal extraction conditions for the extraction of *Auricularia auricula-judae* polysaccharides (AAP) were obtained by using response surface methodology. Furthermore, the aAAP was precipitated from AAP with CTAB, and purified through the DEAE Purose 6 Fast Flow to obtain aAAP-1. The aAAP-1 was acidic polysaccharide with glucuronide and had a synergic anticoagulant activity by the endogenous and exogenous pathway. The senior structure of aAAP-1 needs to be investigated in future work, which will reveal the relationship of the structure–anticoagulant function. AAP should be further explored by using different extraction and purification methods to improve the utilization as a functional food ingredient in antithrombotic health foods, which will have broad market prospects in daily health care and post-disease recovery.

## Figures and Tables

**Figure 1 molecules-25-00710-f001:**
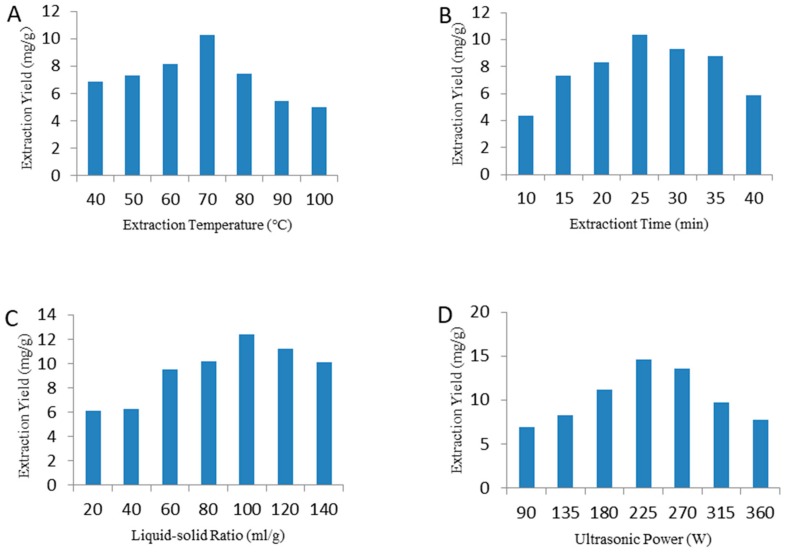
Effects of a different extraction temperature (**A**), extraction time (**B**), liquid-solid ratio (**C**), and ultrasonic power (**D**) on the extraction yield.

**Figure 2 molecules-25-00710-f002:**
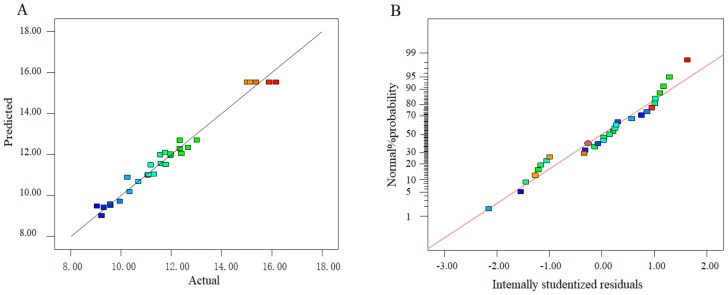
Plots of the adequacy about the proposed model. Plot of the predicted and actual values (**A**); the normal% probability plot (**B**).

**Figure 3 molecules-25-00710-f003:**
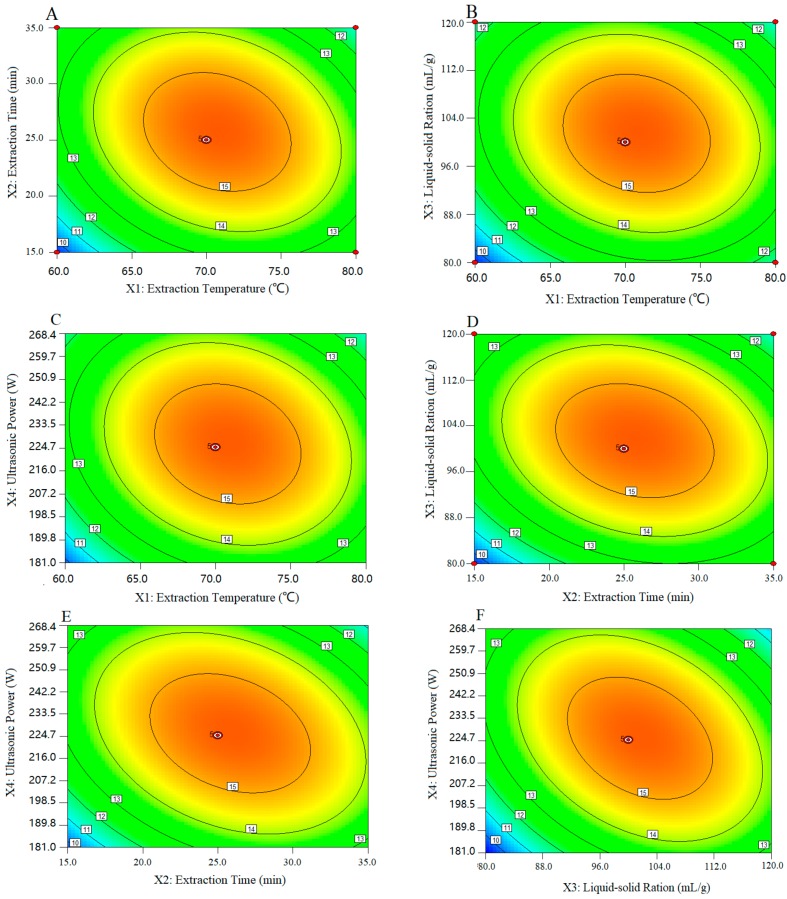
Contour plots showing the significant interactions of the extraction parameters. Extraction temperature and extraction time (**A**), extraction temperature and liquid-solid ratio (**B**), extraction temperature and ultrasonic power (**C**), extraction time and liquid-solid ratio (**D**), extraction time and ultrasonic power (**E**), and liquid-solid ratio and ultrasonic power (**F**).

**Figure 4 molecules-25-00710-f004:**
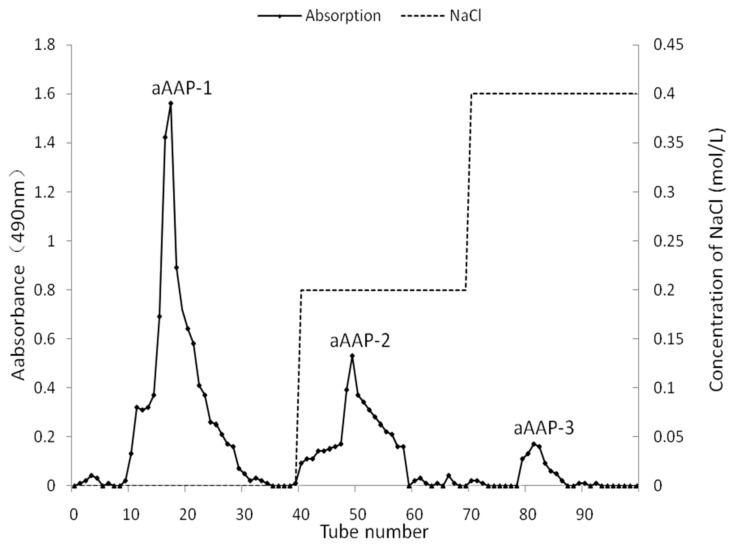
Chromatograms of aAAP-1 separated from cAAPon DEAE Purose 6 Fast Flow.

**Figure 5 molecules-25-00710-f005:**
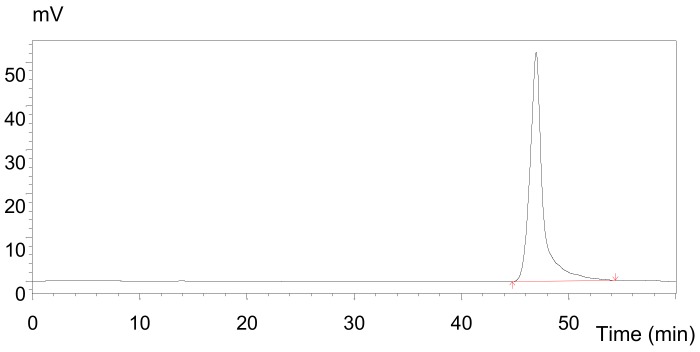
HPLC with tandem gel column profiles of aAAP-1.

**Figure 6 molecules-25-00710-f006:**
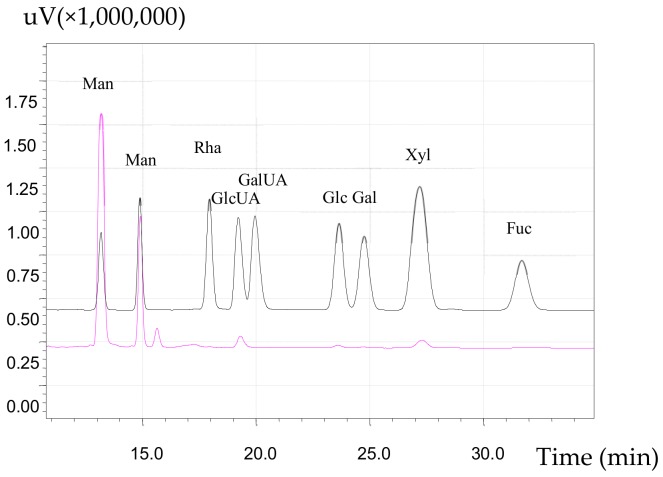
HPLC profiles of aAAP-1 and standard.

**Figure 7 molecules-25-00710-f007:**
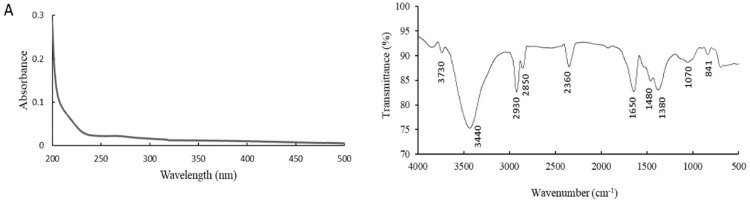
UV (**A**) and FT-IR (**B**) spectrum of aAAP-1.

**Table 1 molecules-25-00710-t001:** Box–Benhnken design and results of AAP extraction experiments.

No.	Codded Variables				Extraction Yield of AAP (mg/g)
	*X*_1_ (Extraction Temperature, °C)	*X*_2_ (Extraction Time, min)	*X*_3_ (Liquid-solid Ratio, ml/g)	*X*_4_ (Ultrasonic Power, W)	
1	0	0	−1	−1	9.23
2	0	0	1	−1	13.02
3	1	0	−1	0	11.79
4	0	0	0	0	15.39
5	−1	−1	0	0	9.59
6	0	−1	0	1	12.67
7	0	1	0	−1	12.34
8	−1	0	−1	0	9.32
9	0	0	0	0	15.02
10	0	0	0	0	16.17
11	0	1	0	1	10.69
12	1	0	1	0	11.09
13	−1	0	1	0	11.19
14	1	1	0	0	11.32
15	0	−1	0	−1	9.57
16	−1	1	0	0	11.97
17	0	−1	1	0	11.76
18	−1	0	0	−1	9.95
19	0	0	1	1	10.34
20	0	−1	−1	0	9.04
21	0	1	−1	0	11.98
22	1	0	0	1	10.26
23	0	0	0	0	15.13
24	1	0	0	−1	11.57
25	0	0	−1	1	12.34
26	0	1	1	0	11.07
27	0	0	0	0	15.9
28	1	−1	0	0	12.41
29	-1	0	0	1	11.59

**Table 2 molecules-25-00710-t002:** The ANOVA results for the response surface quadratic models of AAP yield.

Source	Sum of Squares	DF	Mean Square	*F*-*Value*	*p*-Value
Model	111.82	14	7.99	40.80	<0.0001 **
*X*_1_-extraction temperature	1.94	1	1.94	9.93	0.0071 **
*X*_2_-Extraction time	1.56	1	1.56	7.98	0.0135*
*X*_3_-liquid-solid ratio	1.89	1	1.90	9.68	0.0076 **
*X*_4_-ultrasonic power	0.41	1	0.41	2.08	0.1713
*X* _1_ *X* _2_	3.01	1	3.01	15.38	0.0015 **
*X* _1_ *X* _3_	1.65	1	1.65	8.44	0.0115*
*X* _1_ *X* _4_	2.18	1	2.18	11.12	0.0049 **
*X* _2_ *X* _3_	3.29	1	3.29	16.83	0.0011 **
*X* _2_ *X* _4_	5.64	1	5.64	28.81	<0.0001 **
*X* _3_ *X* _4_	8.38	1	8.38	42.81	<0.0001 **
*X* _1_ ^2^	35.59	1	35.59	181.78	<0.0001 **
*X_2_* ^2^	27.19	1	27.19	138.88	<0.0001 **
*X_3_* ^2^	35.13	1	35.13	179.46	<0.0001 **
*X_4_* ^2^	30.05	1	30.05	153.49	<0.0001 **
Residual	2.74	14	0.20		
Lack of Fit	1.75	10	0.18	0.71	0.6989
Pure Error	0.99	4	0.245		
Cor Total	114.56	28			
*R*^2^ = 0.9761; Adj *R*^2^ = 0.9522; CV = 3.73%; Adeq Precision = 20.454					

* Significant difference (0.01 < *p* < 0.05); ** Extreme significant difference (*p* < 0.01).

**Table 3 molecules-25-00710-t003:** Anticoagulant activity of aAAP-1.

Sample	Concentration (μg/mL)	Clotting Time(s)		
		APTT	PT	TT
Control ^a^		28.0 ± 0.3 ^b^	11.3 ± 0.1	17.7 ± 0.1
Heparin ^c^	2.0	48.7 ± 0.4	67.9 ± 0.2	32.7 ± 0.3
aAAP-1	12.5	26.2 ± 0.3	16.1 ± 0.2 **	20.3 ± 0.1 **
	25.0	27.5 ± 0.4	26.6 ± 0.5 **	21.8 ± 0.1 **
	50.0	47.3 ± 1.7 **	73.7 ± 0.9 **	28.2 ± 0.2 **

* Significant difference (0.01 < *p* < 0.05); **Extreme significant difference (*p* < 0.01). ^a^ Samples were all compared with a blank control (saline); ^b^ Each clotting time were expressed as means ± SD (n = 3); ^c^ the 6th Heparin (196 IU/mg).

**Table 4 molecules-25-00710-t004:** Independent variables and levels in BBD.

Independent Variables	Symbol	Level		
		−1	0	1
Extraction temperature (°C)	*X* _1_	60	70	80
Extraction time (min)	*X* _2_	20	25	30
Liquid-solid ratio (mL/g)	*X* _3_	80	100	120
Ultrasonic power (W)	*X* _4_	180	225	270

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
