# Peer review of "Extraction Optimization, Structural Characterization, and Anticoagulant Activity of Acidic Polysaccharides from Auricularia auricula-judae"

_molecules, 2020, doi:10.3390/molecules25030710_

Round 1
Reviewer 1 Report
This manuscript describes the optimalization of the purification procedure of polysaccharides from the fungus Auricularia Auricula-Judae as acidic but non sulfated polysaccharides endowed with some anticoagulant activity.
Gel filtration revealed a MW of 160'000 daltons and monosaccharide composition analysis showed the presence of Mannose (68%), glucuronic acid (10%), glucose (2%), galactose (1%) and Xylose (31%). It is not sulfated.
The authors suggest that the anticoagulant activity was due to glucuronic acid, despite its low abundance and the lack of sulfation.
The size of the polysaccharide originally extracted was 160 kDa, it was degraded to a homogeneous fraction of 1538 Da by ultrasound treatment.
The purification of the fungus polysaccharide is interesting and data shown are accurate but the text needs extensive rewriting to improve understanding.
It is a very technical paper, that could be improved by showing possible applications for this polysaccharide.
Author Response
Reviewer:
thank you very much for your work, about the English writting, I have revised the draft. Because there were so many revisions so that they were not marked on the manuscript. On this basis, revise and mark your other questions in the manuscript (attachment 1).
Thank you again!
BianChun

Reviewer 2 Report
Dear Authors,
it was my pleasure to review your article titled 'Extraction Optimization, Structural Characterization, and Anticoagulant Activity of Acidic Polysaccharides from Auricularia Auricula'. Overall, it's a good article. The authors correctly selected research methods and described them accordingly. The results obtained were presented correctly. However, some notes in the file should be followed. Please also analyze the entire manuscript for editorial errors. These, however, do not affect the level of work.

Author Response
thanks for your amendments, I have revised and explained them one by one, Please check the attachment for the revised manuscript, the modifications are highlighted in yellow
1.It is necessary to explain all abbreviations used. In its current form, the summary is not fully legible without further delving into the content of the manuscript.
reponse 1: I have explain all the abbreviations used in the first time in my manuscript .
2.Please check and correct the article carefully for editorial errors. In many places the missing spaces and so on. It is impossible to mark all errors.
response2 : all editorial errors Have been modified and marked in the manuscript. Please see the attachment
Please avoid repeating the wording used previously in the title.response3 : Keywords have been adjusted. Please see the attachment
Due to the long name, I suggest using the AAJ abbreviation in the rest of the paper.reponse 4: I consulted the plant taxonomy teacher and will use the abbreviation A. auricula-judae instead of Auricularia auricula-judae in the following articles
This should be combined into one sentence. In its current form, information about production in China has nothing to do with the rest of the sentence.response 5: The sentence has been modified.Please see the attachment
There is no data from statistical analysis. Please complete error bars and homogeneous groups.response 6: It has been modified. Please see the attachment
7.In the current version, the table is difficult to read. The authors have rightly assumed that the data used in BBD should be clarified, but in my opinion it should not be compared with yield results. Here, please show the results for yield and transfer the rest information to supplementary data
response 7: I need to explain the problem. This extraction process requires 29 sets of experiments, so the raw material used in each experiment is 1g, and the amount of extracted polysaccharide is rarely, so the final extracted polysaccharide is not purified, but the amount calculated by the absorption value of phenol - sulfuric acid method. Such optimization experiments cannot take the absorption value as the response, so the yield is used as the response.
8.The charts are too small, or at least the fonts should be enlarged because they are hard to read
response 8: The picture has been modified.Please see the attachment

Reviewer 3 Report
Have a lot of work on your manuscript. But, think that could be much better. See enclosed questions, corrections, suggestions (enclosed file with the text of your manuscript).

Author Response
Thank you very much for your careful and professional advice. I have revised them one by one according to your opinion. Please see the attachment.
Thank you again!
BianChun

Round 2
Reviewer 1 Report
The text has been improved and is is better written but it still needs corrections by an english speaking person.
for example :
line 228
"TT was mainly are flection of the degree of the conversion of fibrinogen into fibrin"
does not make sense.
Reviewer 3 Report
Think it is OK now.
Best wishes!